# Agricultural education in Africa using YouTube multilingual animations: A retrospective feasibility study assessing costs to reach language-diverse populations

**N. Peter Reeves**[1]\*, **Victor Giancarlo Sal y Rosas Celi**[2], **Anne N. Lutomia**[3], **John William Medendorp**[4], **Julia Bello-Bravo**[3], **Barry Pittendrigh**[4]

**1** Sumaq Life LLC, East Lansing, Michigan, United States of America, **2** Departamento de Ciencias, Sección de Matemáticas, Pontificia Universidad Católica del Perú, Lima, Lima, Perú, **3** Department of Agricultural Sciences Education and Communication, Purdue University, West Lafayette, Indiana, United States of America, **4** Department of Entomology, The Urban Center, Purdue University, West Lafayette, Indiana, United States of America

\* reevesn@icloud.com

**Data Availability Statement:** The data is stored on PURR, which is publicly available data repository. A

## Abstract

There is a critical need for widespread information dissemination of agricultural best practices in Africa. Literacy, language and resource barriers often impede such information dissemination. Culturally and linguistically localized, computer-animated training videos placed on YouTube and promoted through paid advertising is a potential tool to help overcome these barriers. The goal of this study is to assess the feasibility of reaching language-diverse populations in Africa using this new type of information dissemination channel. As a case study, cost estimates were obtained for YouTube ad campaigns of a video to prevent postharvest loss through safe food storage using sanitized jerrycan containers. Seventy-three video variants were created for the most common 16 languages in Ghana, 35 languages in Kenya, and 22 languages in Nigeria. Using these videos, campaigns were deployed country wide or focused on zones of influence that represent economically underdeveloped regions known to produce beans suitable for jerrycan storage. Using data collected from YouTube ad campaigns, language-specific models were created for each country to estimate how many viewers could be reached per US dollar spent. Separate models were created to estimate the number of viewers who watched 25% and 75% of the video (most of video without end credits), reflecting different levels of engagement. For language campaigns with both country wide and zone of influence areas of deployment, separate region-specific models were created. Models showed that the estimated number of viewers per dollar spent varied considerably amongst countries and languages. On average, the expected number of viewers per dollar spent were 1.8 (Range = 0.2–7.3) for 25% watched and 0.8 (Range = 0.1–3.2) for 75% watched in Ghana, 1.2 (0.2–4.8) for 25% watched and 0.5 (Range = 0.1–2.0) for 75% watched in Kenya, and 0.4 (Range = 0.2–1.3) for 25% watched and 0.2 (Range = 0.1–0.5) for 75% watched in Nigeria. English versions of the video were the most cost-effective in reaching viewers in Ghana and Nigeria. In Kenya, English language campaigns ranked 28 (country wide) and 36 (zones of influence) out of 37 analyzed campaigns. Results also

link to this repository is provided in the Supplemental information (S2_File).

**Funding:** This work was made possible by the generous support of the American people through the United States Agency for International Development (USAID, https://www.usaid.gov/), under the terms of Contract No. 7200AA20LA00002 (Awardee: Purdue University; PI: BRP). USAID administers the U.S. foreign assistance program providing economic and humanitarian assistance in more than 80 countries worldwide. The contents are the responsibility of the authors and do not necessarily reflect the views of USAID or the United States Government. The funders had no role in study design, data collection and analysis, decision to publish, or preparation of the manuscript.

**Competing interests:** N. Peter Reeves is the Founder and President of Sumaq Life LLC. Sumaq Life LLC applies mathematical modeling approaches to understand complex systems to optimize their performance. It receives funding for these services, including work on the current project. This does not alter our adherence to PLOS ONE policies on sharing data and materials. The remaining authors declare no conflicts of interest in the production of this work.

showed that many local language campaigns performed well, opening the possibility that targeted knowledge dissemination on topics of importance to local populations, is potentially cost effective. In addition, such targeted information dissemination appears feasible, even during regional and global crises when in-person training may not be possible. In summary, leveraging multilingual computer-animations and digital platforms such as YouTube shows promise for conducting large-scale agricultural education campaigns. The findings of the current study provides the justification to pursue a more rigorous prospective study to verify the efficacy of knowledge exchange and societal impact through this form of information dissemination channel.

## Introduction

There is a significant and growing need for widespread information dissemination of agricultural best practices in Africa. For instance, one-third of global food production for human consumption is lost or wasted, totaling 1.3 billion tons per year [1,2], with the primary cause of loss in developing countries being biological spoilage [3]. Given the strong link between agricultural productivity and a country's economic growth [4,5] and population health [6], most governments invest significant funding into large-scale training programs to improve agricultural practices through agricultural extension and advisory services [7,8]. Unfortunately, resources for conducting such training programs are often limited, thus reducing their potential reach and impact. The ratio of agricultural extension agents to farmers are notably high, with estimates revealing ratios of 1 to 1,300 in Ghana [9], 1 to 1,100 in Kenya [10], and even higher ratios in Nigeria, ranging from 1 to 5,000 to as high as 1 to 10,000 [11]. This disparity may, in part, stem from organizational and managerial challenges within extension services, ambiguous extension mandates, and inadequate remuneration for personnel [12]. To address resource constraints, new approaches should be considered for deploying agricultural training programs, to complement traditional in-person agricultural extension training events.

Key impediment for knowledge exchange are literacy and language barriers [13,14]. Although a growing body of scientific knowledge to improve agricultural practices exists [15], translating such knowledge into information that is understandable to and implantable for farmers, often low-literacy and non-English speaking individuals, is challenging [16]. One potential solution for overcoming literacy and language barriers is the use of information and communication technologies (ICTs) in the form of culturally and linguistically localized, computer-animated training videos, specifically developed for low-literacy, non-English speaking audiences [17,18]. Studies suggest such ICTs promote learning gains and adoption of technologies taught in the animations and lead to novel innovations in agricultural communities [19–21]. These prior ICT studies using computer-animated training videos predominately relied on conventional in-person training events. It has been suggested that ICTs could increase access to agricultural extension and advisory services [22–25], thus complementing and enhancing existing extension training programs [26]. However, despite the promise of ICTs, traditional in-person training encounters scalability challenges, primarily stemming from the high ratios of extension agents to farmers.

In this study, the use of a novel approach for information dissemination of ICTs, via YouTube, was explored. YouTube, the online video-sharing platform launched in December 2005, provides open access of content to users. As of 2022, YouTube's global reach has been reported to be +122 million daily active users with services available in more than 100 countries [27].

YouTube ad campaigns, a paid online marketing tool, promote video content across YouTube's user network. The costs of YouTube ad campaigns vary based on audience targeting, bidding, and ad format [28]. Our main goal was to evaluate the feasibility of using YouTube ad campaigns to disseminate information of ICTs to language-diverse populations in Africa, assessing the costs of reaching viewers through this approach. This case study used an educational intervention to address biological spoilage by improving food storage, using Ghana, Kenya, and Nigeria as test countries for Africa.

## Material and methods

This was a retrospective study using data collected as part of the following USAID funded project to address food insecurity in Africa during the Covid-19 pandemic: Scientific Animations Without Borders (SAWBO) Responsive, Adaptive, Participatory Information Dissemination (*RAPID*) Scaling Program (SAWBO-*RAPID*, cooperative agreement number 7200AA20LA00002). The study was deemed to be exempt by Michigan State University's Biomedical and Health Institutional Review Board (IRB number 202101087). The study is not considered human subject research given that it involves the study of existing data that were not specifically collected for research purposes but were generated as part of an information dissemination project. In addition, data provided by YouTube ad analytical reports represent aggregated data, thus recorded in a manner that individual information could not be identified. Given that the study is exempt under 45 CFR 46.101(b), informed consent is not required, nor would be possible with aggregated, de-identified data.

### Online educational intervention

A computer-animated YouTube video was created to educate farmers on how to safely store beans using readily available, sanitized, sealable containers such as jerrycans to avoid post-harvest loss (see the following link to access video https://rapid.sawbo-animations.org/video/1445). Viewers can download the video for free, allowing them to view the videos offline on their mobile device and share the videos with others. This offline usage and propagation of video sharing is not captured in the current study.

### Data collection

Starting July 19 and finishing November 18, 2021, YouTube ad campaigns were conducted to disseminate the jerrycan storage technique across Ghana, Kenya, and Nigeria. Sixteen language variants of the jerrycan video were created in Ghana, 35 in Kenya, and 22 in Nigeria. Campaigns were either run country wide or targeted to specific zones of influence that represent economically underdeveloped areas and regions known to produce beans suitable for jerrycan storage. Zones of influence were defined by USAID Missions in each country and/or by in-country experts. Campaigns were targeted both by geographic location and language (S1.1-S1.3 Tables in S1 File). A cost-per-thousand impressions bidding strategy was used with the average bid set at 9 USD. Impressions represent the number of viewers who watched at least 1 second of the animation [29]. An in-stream skippable ad format was used for all campaigns. Data for the study was obtained from daily YouTube analytical reports generated for each campaign (data link in S2 File, 10.4231/TJK7-YD07), which provided information on money spent, the number of impressions, views (i.e., number of viewers who watched at least 30 seconds of the animation [29]), and the percentage of people who watch 25%, 50%, 75%, and 100% of the jerrycan video. Note that 25% of the video watched provides insight into the number of persons that were exposed to the concept of jerrycan storage, whereas 75% of the

video watched represents the number of people who watched most of the video minus the end credits.

## Variables

Model outputs were the total number of viewers each day who watched 25% and 75% of the jerrycan YouTube video for each campaign. To calculate the number of viewers at the 25% and 75% intervals, the number of views was multiplied by the percentage of people who watched 25% and 75% of the video, respectively. Daily money spent on each campaign, as well as the previous day total number of viewers and money spent were potential factors used to estimate the daily total number of viewers for language-specific models.

## Statistical analysis

An autoregressive distributed lag lineal model was fitted to assess the association between daily number of viewers and money spent. In order to control for autocorrelation between daily observations, four mathematical models with different numbers of input variables were considered (see an example of language-specific modeling in S3 File): first, current daily money spent [Eq 1]; second, current and previous daily money spent [Eq 2]; third, current money spent and previous number of viewers [Eq 3]; and finally, previous and current money spent and previous number of viewers [Eq 4].

$$Y_{(t)} = \beta_2 X_{(t)} + \epsilon_{(t)} \tag{1}$$

$$Y_{(t)} = \beta_2 X_{(t)} + \beta_3 X_{(t-1)} + \epsilon_{(t)} \tag{2}$$

$$Y_{(t)} = \beta_1 Y_{(t-1)} + \beta_2 X_{(t)} + \epsilon_{(t)} \tag{3}$$

$$Y_{(t)} = \beta_1 Y_{(t-1)} + \beta_2 X_{(t)} + \beta_3 X_{(t-1)} + \epsilon_{(t)} \tag{4}$$

where $Y_{(t)}$ is the observed number of viewers on day $t$. $\beta_1$ measures a change in the expected number of viewers following a one viewer increment in the previous day, (t-1); $\beta_2$ measures an immediate change in the expected number of viewers following a one dollar increment in investment in the current day, (t); $\beta_3$ measures a change in the expected number of viewers following a one dollar increment in investment in the previous day, (t-1); and $\epsilon_{(t)}$ is a time series error term. No intercept term was included in the model given that the number of viewers without any money spent is expected to be zero.

To assess the impact of a dollar spent overtime, the long-term effect on the number of viewers was estimated. For instance, one dollar increments in investment at a given time are associated with a $\beta_1\beta_2 + \beta_3$ estimated daily number of viewers. Furthermore, imposing the stability condition, $|\beta_1| < 1$, $(\beta_2 + \beta_3)/(1 - \beta_1)$, representing the long-term effect, is the expected change on the average number of viewers across all days following a one-dollar increment in money spent. For models captured using Eqs 1–3, parameters not included are set to zero.

Fig 1 shows a graphic representation of the presented models. The contribution of previous views could be thought of as a priming effect, given that higher previous views increased the current day number of viewers. The contribution of previous costs, on the other hand, could be interpreted as a saturation effect, given that the higher the previous cost the fewer current day viewers tended to be reached.

The Durbin-Watson test was applied to the estimated time series error, $\epsilon_{(t)}$, to assess evidence of autocorrelation (i.e., residual errors amongst days are linear dependent). Final models

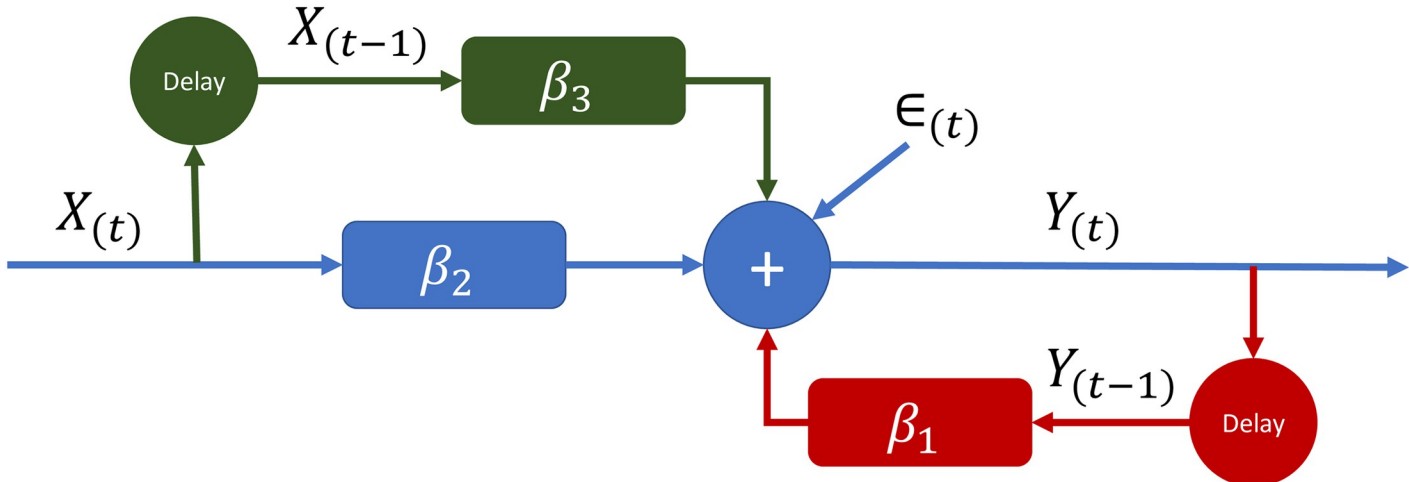

**Fig 1. Schematic of YouTube ad campaign models.** The model estimates number of viewers reached, $Y_{(t)}$ on day t. Note that the model can use past information, such as previous views, $Y_{(t-1)}$, and previous costs, $X_{(t-1)}$, to improve the estimation of the number of viewers. The delay (t-1) represents a delay of 1 day and $\epsilon_{(t)}$ represents an error term that accounts for the unexplained variability in the data on day $t$. The simplest model ($Y_{(t)} = \beta_2 X_{(t)} + \epsilon_{(t)}$) is captured by the blue pathway. The more complex model ($Y_{(t)} = \beta_2 X_{(t)} + \beta_3 X_{(t-1)} + \epsilon_{(t)}$) is captured by the blue and green pathways. The more complex model ($Y_{(t)} = \beta_1 Y_{(t-1)} + \beta_2 X_{(t)} + \epsilon_{(t)}$) is captured by the blue and red pathways. Finally, the most complex model ($Y_{(t)} = \beta_1 Y_{(t-1)} + \beta_2 X_{(t)} + \beta_3 X_{(t-1)} + \epsilon_{(t)}$), and the one most used, is captured by the blue, green, and red pathways.

that violated autocorrelation are reported and discussed. The Akaike information criterion (AIC) was considered for model comparison to select the final model. For the final model, confidence intervals for the long-term effect were computed using maximum entropy bootstrap approach that can provide reliable estimates even under non-stationary conditions [30].

For each of the three African countries, language-specific models were created to estimate the expected number of viewers per dollar spent. For language campaigns with both regions of deployment (i.e., country wide and zone of influence), separate region-specific models were created. To ensure a sufficient number of observations for model estimation, only campaigns that lasted longer than 50 days were used to generate models. Excluded campaigns in Ghana were for language variants in Sisaali (9 days) and Bono (13 days), and in Nigeria, language variants in Abi (7 days), Isoko (7 days), Itshekiri (13 days), Kalabari (7 days), Urhobo (7 days), and Yakkur (13 days).

All statistical analyses were conducted with R version 4.1.3 [31].

## Results and discussion

In total, 36,271,249 impressions were generated through 7,889 YouTube ad campaigns promoting the jerrycan animation (Table 1). Across all countries, YouTube ad campaigns lead to 149,131 viewers who were exposed to the jerrycan technique (i.e., 25% of video watched) and 63,952 viewers who watched 75% of the video. There was considerable variability in terms of money spent across languages as well as across countries, with the most money spent in Kenya (median cost of campaigns = \$1,639), followed by Ghana (median cost of campaigns = \$867), and then Nigeria (median cost of campaigns = \$615). Furthermore, there was variability on the campaign duration, with campaigns in Kenya having a median duration of 116 days, while Ghana and Nigeria had median durations of 102 and 101 days, respectively. Campaigns in Kenya had the most viewers, 3.9 times of that of Ghana, and 7.6 times more than Nigeria. The median daily amount of money spent on a campaign was \$9.39, \$7.13, and \$7.14 in Kenya,

**Table 1. Observed total number of campaigns and viewers for Ghana, Kenya, and Nigeria.**

| Country | Number of daily campaigns | Impressions[a] | Views[b] | Total number of viewers who watched 25% of video | Total number of viewers who watched 75% of video |
|---|---|---|---|---|---|
| Ghana | 1,632 | 5,209,652 | 692,105 | 35,615 | 15,785 |
| Kenya | 4,454 | 27,009,253 | 2,712,361 | 104,933 | 44,711 |
| Nigeria | 1,803 | 4,052,344 | 356,486 | 8,583 | 3,456 |
| Total | 7,889 | 36,271,249 | 3,760,952 | 149,131 | 63,952 |

[a] Impressions represent the number of times the jerrycan animation was shown to YouTube viewers, with a view lasting more than 1 second.

[b] Views represent the number of YouTube viewers who watched 30 seconds or more of the video.

Ghana, and Nigeria, respectively (Table 2). Daily trends followed overall trends with Kenyan campaigns having higher viewership than Ghana campaigns (25% daily average: 24.0 vs. 21.8; 75% daily average: 10.1 vs. 9.7) and significantly more that of Nigerian campaigns (25% daily average: 24.0 vs. 4.8; 75% daily average: 10.1 vs. 1.9).

The most frequent language-specific model type used to predict viewership was the one that included current money spent, previous views, and previous costs. Out of 69 language-specific models for predicting 25% watched, 59 (85.5%) showed no evidence of autocorrelation among the estimated residuals based on the Durbin-Watson test. Out of these, 49 were best represented by the three input variables, while models for 8 languages (Luhya-Idakho, Fante, Nzema, Gonja, Edo, Dangme, Hausa, and Pokot) relied solely on the previous number of viewers and current cost. Additionally, the Kasem and Efik language models performed optimally with only current and previous cost as explanatory variables. Among the models that satisfied the non-autocorrelation condition, the average adjusted $R^2$ was 0.93 (Range 0.65 to 0.98).

In terms of models predicting 75% watched, 57 (82.6%) showed no evidence of autocorrelation. Among those, 48 were described with the three-input variable model, six (Luhya-Idakho, Giryama, Dangme, Gonja, Fante, and Ga) with previous number of viewers and current cost, one (Kasem) with current and previous cost, and two (Efik and Pokot) with only current cost. Overall, these non-autocorrelated models also explained a significant amount of the variance in the data with the combined average adjusted $R^2$ of 0.90 (Range 0.60 to 0.98).

In Ghana, the strongest long-term effect was estimated in the English country wide campaign for both the 25% and 75% watched models (25% watched: 7.3 viewers per USD [95%CI: 5.5 to 8.1]; 75% watched: 3.2 viewers per USD [95%CI: 2.5 to 3.7], see Figs 2 and 3). In Kenya, the strongest long-term effect was estimated in the Kikuyu campaign for both the 25% and 75% watched models (25% watched: 4.8 viewers per USD [95%CI: 3.5 to 5.6]; 75% watched: 2.0 viewers per USD [95%CI: 1.5 to 2.4]). In Nigeria, the strongest long-term effect was estimated in the English country wide campaign for both the 25% and 75% watched models (25% watched: 1.3 viewers per USD [95%CI: 1.0 to 1.4]; 75% watched: 0.5 viewers per USD [95%CI: 0.4 to 0.6]). The second and third strongest long-term effects were observed with English zone of influence (75% watched: 0.4 viewers per USD [95%CI: 0.3 to 0.4]) and Pidgin (75% watched: 0.3 viewers per USD [95%CI: 0.2 to 0.3]) campaigns. On average, the long-term expected change in number of viewers per dollar spent were 1.8 (watched 25%) and 0.8 (watched 75%) in Ghana, 1.2 (watched 25%) and 0.5 (watched 75%) in Kenya, and 0.4 (25%) and 0.2 (75%) in Nigeria.

The goal of this study was to assess the feasibility of reaching language-diverse populations in Africa using YouTube ad campaigns, to help bridge gaps in access to agricultural extension and advisory services. The results suggest that YouTube ad campaigns have the potential to reach large, language-diverse audiences. The cost to reach viewers varied considerably between

**Table 2. Observed descriptive statistics reflecting the daily performance of YouTube ad campaigns in Ghana, Kenya, Nigeria, and combined.**

| | | Ghana | Kenya | Nigeria | Combined |
|---|---|---|---|---|---|
| Daily money spent per campaign (USD) | Mean | $9.26 | $19.15 | $8.76 | $14.54 |
| | Standard deviation | $15.97 | $37.26 | $15.84 | $29.97 |
| | Median | $7.13 | $9.39 | $7.14 | $7.19 |
| | Interquartile | $1.11 –$7.26 | $1.27 –$21.58 | $0.93 –$7.27 | $1.12 –$21.48 |
| | Range | $0.02 –$159.90 | $0.01 –$446.30 | $0.02 –$158.16 | $0.01 –$446.30 |
| Daily number of viewers per campaign who watched 25% of video | Mean | 21.8 | 24 | 4.8 | 18.9 |
| | Standard deviation | 84.9 | 51.6 | 15.5 | 56 |
| | Median | 5.2 | 8.9 | 1.3 | 4.7 |
| | Interquartile | 1.9–13.7 | 2.4–18.7 | 0.4–2.8 | 1.3–14.4 |
| | Range | 0–1533.9 | 0–747.5 | 0–205.9 | 0–1533.9 |
| Daily number of viewers per campaign who watched 75% of video | Mean | 9.7 | 10.1 | 1.9 | 8.1 |
| | Standard deviation | 38.9 | 23.1 | 6.4 | 25.3 |
| | Median | 2.1 | 3.7 | 0.5 | 1.9 |
| | Interquartile | 0.8–5.8 | 1.0–8.1 | 0.1–1.1 | 0.5–5.9 |
| | Range | 0–747.2 | 0–376.0 | 0–89.9 | 0–747.2 |

countries and languages. In general, Ghana and Kenya campaigns tended to have lower costs to reach viewers than Nigeria campaigns. In addition, English campaigns were the most cost-effective in Ghana and Nigeria; however, in Kenya, many of the local language campaigns were more cost effective. And not surprisingly, pursuing higher levels of engagement increased the cost for online information dissemination (on average, $0.78 to have one viewer watch 25% of video versus $1.78 to have one viewer watch 75% of video). Results also show that linking campaign spending to viewership could have improved the overall project cost-effectiveness: some underfunded, but well-performing campaigns (e.g., Ga in Ghana) could have receive additional support to increase overall viewership (see Figs 2 and 3). This suggests that future campaigns would benefit by adopting more formal approaches such as real-time analysis, machine learning, and optimization theory to maximize information dissemination investments.

In terms of study limitations, it is not known if only the intended audience, which in this case includes, but is not limited to, African farmers, was reached. It is possible that other groups not directly associated with farming also comprised the viewership. Not all farmers have devices or internet/cellular coverage, particularly in rural areas. However, in many African countries, the divide between urban and rural is not well defined, with urban populations also owning and maintaining farms [32,33] and communication between family members and clans from urban to rural areas being more fluid than in more developed Western economies [34]. In terms of impact, the current study was not designed to assess whether viewing online training videos translates to adoption of the technique, and whether this then leads to positive societal impact (e.g., reducing post-harvest loss and food insecurity). There is evidence to support that use of computer-animated videos as a training tool led to upwards of an 89% level of adoption [20,21], when the videos were part of an in-person extension intervention. However, it remains to be determined the adoption rates that occur through large-scale YouTube information dissemination campaigns. Another limitation related to the modelling is that a small portion of the models did not meet the non-autocorrelation assumption. For those models that violated this assumption, the maximum entropy bootstrap approach applied has been shown to provide reliable interval estimates [30].

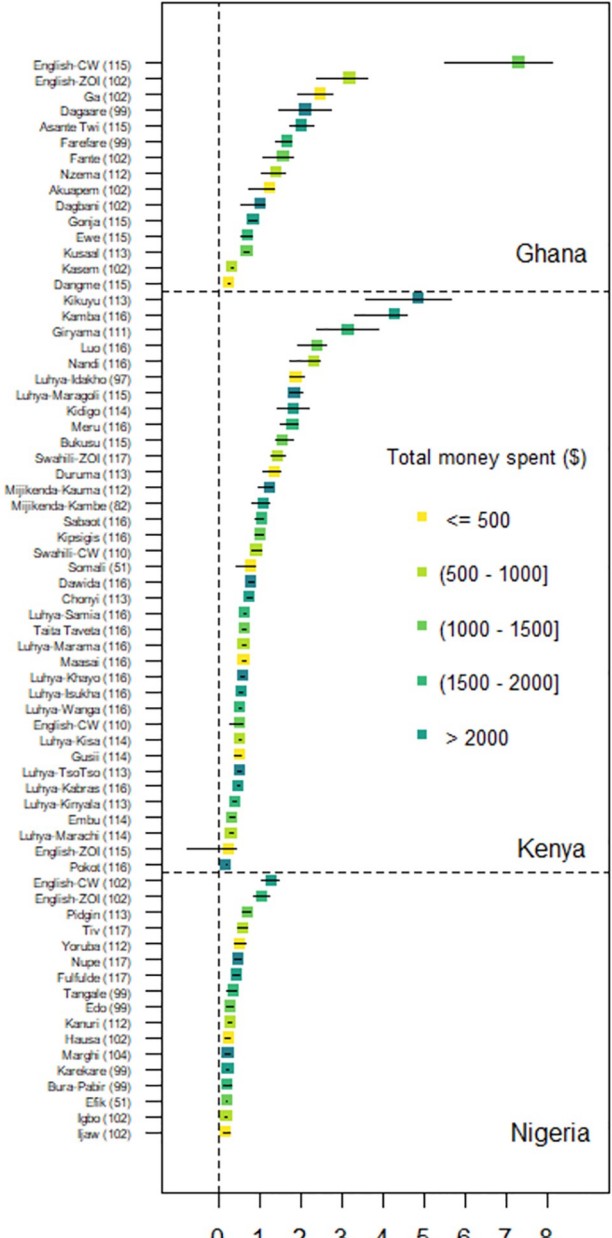

**Fig 2. Expected number of viewers who watched 25% of the video.** YouTube ad campaigns shown on y-axis (# of daily campaigns) with long-term effect point estimates of expected number of viewers who watched 25% of the video with confidence intervals for Ghana (top), Kenya (middle), and Nigeria (bottom). The dotted vertical line represents a long-term effect of zero, meaning that the campaign is expected to reach no viewers. CW refers to country wide campaign and ZOI refers to zone of influence campaign.

Factors that contribute to the cost of a campaign are targeting, bidding, and ad format (e.g., in-stream skippable, in-stream non-skippable) [28]. The ad format was consistent amongst campaigns, suggesting that targeting and bidding resulted in cost differences. Assuming supply and demand dynamics exist for online campaigns, one would expect that targeting a larger

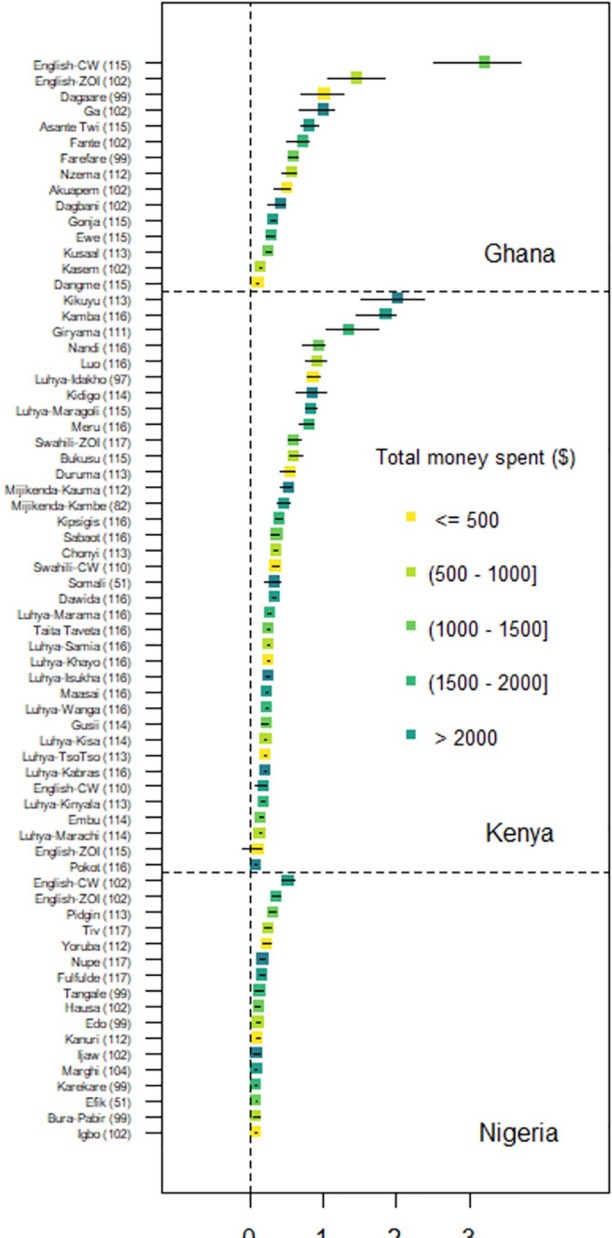

**Fig 3. Expected number of viewers who watched 75% of the video.** YouTube ad campaigns shown on y-axis (# of daily campaigns) with long-term effect point estimates of expected number of viewers who watched 75% of the video with confidence intervals for Ghana (top), Kenya (middle), and Nigeria (bottom). The dotted vertical line represents a long-term effect of zero, meaning that the campaign is expected to reach no viewers. CW refers to country wide campaign and ZOI refers to zone of influence campaign.

audience would reduce costs. There is evidence to support this as English country wide campaigns were more cost-effective than English zone of influence campaigns in all three countries. However, one would also expect that Nigeria with its larger population would have more cost-effective campaigns than Ghana and Kenya, but this does not appear to be the case. One

possible explanation for the higher costs in Nigeria may be higher bidding. Possibly, there is more competition for YouTube advertising space in Nigeria than in the other two countries, which could be driving up the cost per viewer.

In the context of model insights, using previous-day information improved viewer estimate accuracy in most campaigns. The previous viewer factor, $\beta_1$, provides insight into when campaigns are more likely to propagate, with more effective campaigns having a larger $\beta_1$. Among the non-autocorrelated three-variable models predicting 75% watched, $\beta_1$ values were consistently less than 1 (Range = 0.17 to 0.90, IQR = 0.52 to 0.70). Although not observed in this study, $\beta_1 > 1$ would reflect conditions in which a video would go "viral", which in this context is desirable. The previous cost factor, $\beta_3$, on the other hand, may indicate when a campaign is becoming saturated. $\beta_3$ was negative, which suggests that previous money spent on a campaign will reduce the expected current number of viewers. Such a situation could occur if the same viewers were being targeted in consecutive days, and the viewers were not inclined to view the animation again. The average estimated value of $\beta_3$ across these campaigns was -0.2 (Range = -1.8 to 0, IQR = -0.2 to -0.1), suggesting that some campaign saturation may have occurred. Given that models are still early in development, future studies should explore in more detail the time-varying properties of campaigns, and if warranted, use adaptive modeling to provide more precise estimates of cost-effectiveness, and explore priming and saturation changes with time. Such information may be important for optimizing campaigns in real-time by selecting languages and regions with the lowest cost per viewer.

Online agricultural education using YouTube ads could be a cost-effective method for reaching large populations. In comparison, in-person agricultural extension training in Ghana using videos as a training tool has been estimated to be $78 per person [35]; while farmer field schools in Ghana range from $8 to $10 per person and in East Africa vary between $9 to $35 per person [36]. In contrast, the observed YouTube ad costs to have a viewer watch 75% of video, averaged across all campaigns, was $0.94 in Ghana, $4.4 in Nigeria, and $1.89 in Kenya. It is important to note that these videos are free to download and share with others. The cost per viewer does not account for information sharing so cost estimates may be lower than reported. However, further research is required to confirm that the intended audience was reached, and that knowledge gain occurred in viewers.

Besides the potential cost effectiveness of YouTube dissemination, an important benefit of online education versus in-person training is the ability to reach end-users during a crisis. One of the impetuses for the SAWBO *RAPID* project was to address food insecurity caused from the Covid-19 pandemic. During the COVID-19 pandemic, digital learning emerged as a critical tool in Africa to mitigate the disruption caused by government policies limiting in-person teaching [37]. However, other crises, such as regional security concerns, like those in North Eastern Kenya [38] and Northern Nigeria [39], that make in-person training difficult to conduct, represent other circumstances where YouTube information dissemination may be a logical option.

## Conclusions

This feasibility study suggests that online information dissemination through YouTube ads can reach significant numbers of language, diverse viewers in Africa.

By demonstrating feasibility, prospective intervention studies are warranted to confirm that videos reach their intended audience, viewing videos lead to adoption of techniques, and adoption of techniques translate to tangible societal benefits. If subsequent studies demonstrate impactful use of educational content, it may be imperative for donor agencies to develop working relationships with online platforms like YouTube to develop "bulk" costing structures that

may drive down these costs further. Such collaborations could enable these platforms to become effective information dissemination channels in international development, especially in times of crisis be they regional or global.

## Supporting information

**S1 File. YouTube Ad campaigns.** Links to YubeTube multilingual animations and geographical locations targeted in YouTube Ad campaigns.
(DOCX)

**S2 File. Data link.** YouTube data used to generate the language-specific models.
(DOCX)

**S3 File. Language-specific model example.** Model details for language-specific model generated from YouTube data for daily campaigns in Ghana for the language English (zone of influence) and considering number of viewers that watched 75% of the video.
(DOCX)

## Author Contributions

**Conceptualization:** N. Peter Reeves, Victor Giancarlo Sal y Rosas Celi, Anne N. Lutomia, John William Medendorp, Julia Bello-Bravo, Barry Pittendrigh.

**Data curation:** N. Peter Reeves.

**Formal analysis:** N. Peter Reeves, Victor Giancarlo Sal y Rosas Celi.

**Funding acquisition:** Julia Bello-Bravo, Barry Pittendrigh.

**Methodology:** N. Peter Reeves, Victor Giancarlo Sal y Rosas Celi, Anne N. Lutomia, John William Medendorp, Julia Bello-Bravo, Barry Pittendrigh.

**Project administration:** John William Medendorp, Barry Pittendrigh.

**Supervision:** Barry Pittendrigh.

**Visualization:** N. Peter Reeves, Victor Giancarlo Sal y Rosas Celi, Anne N. Lutomia, John William Medendorp, Julia Bello-Bravo, Barry Pittendrigh.

**Writing – original draft:** N. Peter Reeves, Victor Giancarlo Sal y Rosas Celi, Barry Pittendrigh.

**Writing – review & editing:** N. Peter Reeves, Victor Giancarlo Sal y Rosas Celi, Anne N. Lutomia, John William Medendorp, Julia Bello-Bravo, Barry Pittendrigh.

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
