## [Decision Letter · Decision Letter 0]

12 Jan 2024

PONE-D-23-30519Language-diverse agricultural education in Africa using YouTube Ad campaigns: A retrospective feasibility studyPLOS ONE

Dear Dr. Reeves,

Thank you for submitting your manuscript to PLOS ONE. After careful consideration, we feel that it has merit but does not fully meet PLOS ONE’s publication criteria as it currently stands. Therefore, we invite you to submit a revised version of the manuscript that addresses the points raised during the review process.

I appreciate the overall quality and relevance of your work and consider that the manuscript is well-crafted. Two reviewers have provided valuable feedback, suggesting some minor improvements such as clarifying the methodology and checking statistical analyses. I understand that the recommendation to improve the measurement of the knowledge increment obtained by farmers goes beyond the scope of the article's objectives. Nevertheless, the authors may consider including some additional reflections on this topic for future consideration, in addition to those already provided in the discussion. I kindly request that you review the reviewers' suggestions and carefully consider how to incorporate them into the article if you believe they contribute to its quality. Alternatively, please provide justification if you choose not to implement specific recommendations.

We look forward to receiving your revised manuscript.

Kind regards,

Alfonso Rosa Garcia

Academic Editor

PLOS ONE

“This work was made possible by the generous support of the American people through the United States Agency for International Development (USAID,

https://www.usaid.gov/), under the terms of Contract No. 7200AA20LA00002 (Awardee: Purdue University; PI: BRP). USAID administers the U.S. foreign assistance

program providing economic and humanitarian assistance in more than 80 countries worldwide. The contents are the responsibility of the authors and do not necessarily reflect the views of USAID or the United States Government.”

“N. Peter Reeves is the Founder and President of Sumaq Life LLC. Sumaq Life LLC applies mathematical modeling approaches to understand complex systems to optimize their performance. It receives funding for these services, including work on the current project. The remaining authors declare no conflicts of interest in the production of this work.”

6. We note that Figure 1 in your submission contain copyrighted images. All PLOS content is published under the Creative Commons Attribution License (CC BY 4.0), which means that the manuscript, images, and Supporting Information files will be freely available online, and any third party is permitted to access, download, copy, distribute, and use these materials in any way, even commercially, with proper attribution. For more information, see our copyright guidelines: http://journals.plos.org/plosone/s/licenses-and-copyright.

Reviewers' comments:

Reviewer's Responses to Questions

**Comments to the Author**

1. Is the manuscript technically sound, and do the data support the conclusions?

Reviewer #1: Yes

Reviewer #2: Partly

2. Has the statistical analysis been performed appropriately and rigorously? 

Reviewer #1: Yes

Reviewer #2: No

3. Have the authors made all data underlying the findings in their manuscript fully available?

Reviewer #1: Yes

Reviewer #2: Yes

4. Is the manuscript presented in an intelligible fashion and written in standard English?

Reviewer #1: Yes

Reviewer #2: Yes

5. Review Comments to the Author

Reviewer #1: REVIEWERS’ COMMENTS: PONE-D-23-30519

1. The study presents the results of original research.

Yes, the research results are original.

2. Results reported have not been published elsewhere.

The reviewer could not find the results in other publications. However, that may not sufficient because the reviewer’s role was limited in this regard. Therefore, the authors should declare to the Journal that the results have not been published elsewhere.

3. Experiments, statistics, and other analyses are performed to a high technical standard and are described in sufficient detail.

The research methods and statistical analyses meet the requirements for a research manuscript.

4. Conclusions are presented in an appropriate fashion and are supported by the data.

The conclusions are based on the research findings. However, they should be separated from the discussions.

5. The article is presented in an intelligible fashion and is written in standard English.

The article is written in standard English.

6. The research meets all applicable standards for the ethics of experimentation and research

integrity.

Yes, the research meets all applicable standards for the ethics of experimentation and research integrity.

7. The article adheres to appropriate reporting guidelines and community standards for data

availability.

The primary data was made available by the authors.

DETAILED REVIEWER’S COMMENTS

1. Abstract

Add the summary of the recommendations in the abstract. Food insecurity should be removed from key words because it was not covered by the study.

2. Introduction

The introduction is well written with flow of ideas. Again, the research problem is clearly articulated. The following improvements should be made in the introduction.

• Provide the information for farmer to extension ration for countries that participated in the study (Ghana, Kenya and Nigeria) to supplement the ratio for sub-Saharan Africa. Again, indicate the reasons why the farmer to extension ratio is a high in sub-Saharan Africa or selected countries.

• Agricultural extension advisory services sho

---

## [Author Response · Author response to Decision Letter 0]

16 Feb 2024

Please see Response to reviewer document included in the resubmission.

---

## [Editor Report · Decision Letter 1]

28 Mar 2024

Agricultural education in Africa using YouTube multilingual animations: A retrospective feasibility study assessing costs to reach language-diverse populations

PONE-D-23-30519R1

Dear Dr. Reeves,

We’re pleased to inform you that your manuscript has been judged scientifically suitable for publication and will be formally accepted for publication once it meets all outstanding technical requirements.

Kind regards,

Alfonso Rosa Garcia

Academic Editor

PLOS ONE
---

## [Editor Report · Acceptance letter]

8 Apr 2024

PONE-D-23-30519R1 

PLOS ONE

Dear Dr. Reeves, 

I'm pleased to inform you that your manuscript has been deemed suitable for publication in PLOS ONE. Congratulations! Your manuscript is now being handed over to our production team.

Kind regards, 

on behalf of

Dr. Alfonso Rosa Garcia 

Academic Editor

PLOS ONE